# Assessing the quality of CKD care using process quality indicators: A scoping review

Na Zhou[1,2☯], Chengchuan Chen[3☯], Yubei Liu[1], Zhaolan Yu[4], Aminu K. Bello[5], Yanhua Chen[1,6]*, Ping Liu[1,7]*

1 School of Nursing, Southwest Medical University, Luzhou, Sichuan Province, China, 2 Department of Cardiology, The First Affiliated Hospital, Chengdu Medical College, Chengdu, Sichuan Province, China, 3 Department of Anesthesiology, Chengdu Qingbaijiang District Traditional Chinese Medicine Hospital, Chengdu, Sichuan Province, China, 4 Department of Nephrology, The Affiliated Hospital of Southwest Medical University, Luzhou, Sichuan Province, China, 5 Faculty of Medicine and Dentistry, Division of Nephrology and Immunology, University of Alberta, Edmonton, Alberta, Canada, 6 Department of Nursing, The Affiliated Hospital of Southwest Medical University, Luzhou, Sichuan Province, China, 7 Departments of Medicine and Community Health Sciences, Cumming School of Medicine, University of Calgary, Calgary, Alberta, Canada

☯ These authors contributed equally to this work.
* chen_yanhua25@163.com (YC); ping.liu1@ucalgary.ca (PL)

**Data Availability Statement:** All relevant data are within the manuscript and its Supporting information files.

## Abstract

### Introduction

Assessing the quality of chronic kidney disease (CKD) management is crucial for optimal care and identifying care gaps. It is largely unknown which quality indicators have been widely used and the potential variations in the quality of CKD care. We sought to summarize process quality indicators for CKD and assess the quality of CKD care.

### Methods

We searched databases including Medline (Ovid), PubMed, Cochrane Library, Web of Science, CINAHL, and Scopus from inception to June 20, 2024. Two reviewers screened the identified records, extracted relevant data, and classified categories and themes of quality indicators.

### Results

We included 24 studies, extracted 30 quality indicators, and classified them into three categories with nine themes. The three categories included laboratory measures and monitoring of CKD progression and/or complications (monitoring of kidney markers, CKD mineral and bone disorder, anemia and malnutrition, electrolytes, and volume), use of guideline-recommended therapeutic agents (use of medications), and attainment of therapeutic targets (blood pressure, glycemia, and lipids). Among the frequently reported quality indicators (in five or more studies), the following have a median proportion of study participants achieving that quality indicator exceeding 50%: monitoring of kidney markers (Scr/eGFR), use of medications (ACEIs/ARBs, avoiding non-steroidal anti-inflammatory drugs (NSAIDs)), management of blood pressure (with a target of ≤140/90, or without specific targets), and

**Funding:** Financial Disclosure: PL was supported by a Kidney Research Scientist Core Education and National Training (KRESCENT) Program New Investigator Award (FRN 2023KNIA-1058404), which is co-sponsored by the Kidney Foundation of Canada and Canadian Institutes of Health Research).

**Competing interests:** The authors have declared that no competing interests exist.

monitoring for glycated hemoglobin A1c (HbA1c)). The presence of diabetes, hypertension, cardiovascular disease, or proteinuria was associated with higher achievement in indicators of monitoring of kidney markers, use of recommended medications, and management of blood pressure and glycemia.

## Conclusion

The quality of CKD management varies with quality indicators. A more consistent and complete reporting of key quality indicators is needed for future studies assessing CKD care quality.

## Introduction

Chronic kidney disease (CKD) is a significant public health concern, with a global prevalence of approximately 9.1% [1]. CKD is defined as abnormalities of kidney structure or function, present for more than 3 months, including the presence of markers of kidney damage (e.g., albuminuria) or a persistent decrease in glomerular filtration rate (GFR) <60 ml/min/1.73 m$^2$ [2]. CKD is associated with increased risks of cardiovascular disease, acute kidney injury, kidney failure, and mortality [3]. Risk factors for CKD progression are multifaceted, including advanced age, diabetes mellitus, hypertension, the presence of proteinuria, lifestyle, and other related factors [4,5]. Optimal CKD management may slow the progression of the disease and reduce CKD-related morbidity and mortality [6–8]. Assessing the quality of CKD management is a crucial step toward optimal patient care, and it also provides valuable feedback for quality assurance and improvement programs [9,10].

Quality indicators can be used to measure the quality of CKD management [11]. National and international clinical practice guidelines for CKD management recommend several quality indicators, including blood pressure control, lipid management, and appropriate use of medications such as angiotensin-converting enzyme inhibitors (ACEIs) or angiotensin II receptor blockers (ARBs) [2,12]. Many other quality indicators for CKD have also been proposed, refined, and updated, and studies have applied various quality indicators to assess the quality of CKD care [13–15]. While there is a systematic review of the validity of process quality indicators for CKD [9], it is largely unknown what quality indicators have been widely applied in research studies, and the current status of, and potential variations in, the quality of CKD care.

We conducted a scoping review to summarize process quality indicators for CKD management and assess the quality of CKD care using the identified quality indicators. Such a review will be useful for understanding a set of common quality indicators for CKD management, measuring adherence to guidelines, and identifying areas for improvement.

## Methods

We reported this scoping review according to the Preferred Reporting Items for Systematic Reviews and Meta-Analyses extension for Scoping Reviews (PRISMA-ScR) (S1 File) [16]. A scoping review is a literature review method that helps identify knowledge gaps, evaluate the extent and nature of research on a particular topic, and synthesize evidence from various sources [17]. We chose this approach because there is potentially a diverse body of literature assessing quality indicators for CKD management, and there is a lack of standardized quality

of care assessment methodologies and complete reporting. Our review protocol was registered with the OpenScience Framework (osf.io/4h3tw) on 18 February 2023.

## Eligibility criteria

We included original studies in this review if they assessed the quality of CKD management by applying process quality indicators in adults (18 years of age or older) with CKD. The following studies were excluded:

1. Studies in people with kidney failure with or without kidney replacement therapy, acute kidney injury, or acute kidney disease.

2. Focusing on developing process quality indicators for CKD management.

3. Assessing the associations between quality indicators and patient outcomes (such as survival, cardiovascular events, kidney failure, and quality of life).

4. Focusing on early identification of CKD.

5. Developing quality indicators for conservative management of advanced CKD.

6. Reviews, editorials, commentaries, and recommendations on CKD management.

## Search and selection strategies

We searched studies in Medline (Ovid), PubMed, Cochrane Library, Web of Science (Core Collection), CINAHL, and Scopus from inception to June 20, 2024, using the terms "quality of care/quality indicators" and "chronic kidney disease" (see search strategy in S2 File). Database searches were restricted to English only. We carried out additional hand searches by tracking citations and references of included studies. We reviewed major guidelines for management of CKD, including KDIGO and those from UK [18], US [19], Canada [20], and China [21]. Two reviewers (NZ and CCC) independently conducted two phases of the screening process (title/abstract screening and full-text screening). In case of doubt, a third reviewer (PL) was involved. NoteExpress software (version 3.7.0) was used for screening the identified articles.

## Data extraction and analysis

We extracted data from all included studies using a standardized data extraction form. General study characteristics for each study included author, publication year, country, study design, setting, data source, participant recruitment, inclusion and exclusion criteria, participant characteristics (CKD diagnosis and eGFR categories), and study sample size. In addition, we identified sources of quality indicators or methods for developing quality indicators (the Delphi process, specific guidelines, multiple sources, or unclear sources).

We extracted quality indicators of included studies and grouped them into three categories with nine themes according to previous studies and the recommendations from guidelines [2,12]. To assess the quality of CKD management, we extracted the quality indicators and listed the common quality indicators that were reported in at least five studies. We extracted the proportion of the study population reported for common quality indicators of original studies, then used the median and interquartile range (IQR) to summarize the distribution of proportions of study participants meeting a quality indicator of included studies. We reported the proportions of patients meeting a quality indicator by comorbid status, including diabetes,

hypertension, cardiovascular disease, and proteinuria. Missing data were excluded from the statistical analysis.

### Risk of bias assessment or quality appraisal

Following guidance on scoping review conduct [16], we did not perform a risk of bias assessment or quality appraisal for the included studies.

### Patient and public involvement

The patients and the public were not involved in this scoping review.

## Results

### Study inclusion

We identified 12,595 studies from six databases and additional three from hand searches. We removed 3,134 duplicates and excluded 9,428 studies after the title and abstract screening. We assessed 36 studies with full-text for eligibility. Of them, 24 studies [13–15,22–42] were included in this review (S1 Fig).

### Characteristics of included studies

The included studies were from 12 different countries, 10 studies from North America, followed by Europe, Asia, and Oceania. Most studies were conducted in CKD G3 to G5 (50%), in a primary care setting (62.5%), and with a cohort study design (70.8%). In terms of sources of quality indicators, 13 studies referred to international or national guidelines; five studies did not report the sources of quality indicators; four studies developed quality indicators through collecting opinions from a group of experts in the relevant field (the Delphi process); two studies identified quality indicators through multiple sources (Table 1 in S1 Table).

### Extracted quality indicators and relevant themes

Overall, 30 quality indicators were extracted from the included studies. These quality indicators were classified into three categories with nine themes (Table 2 in S2 Table):

A. Laboratory measures and monitoring of CKD progression and/or complications

- Monitoring of kidney markers[2]

- CKD mineral and bone disorder (CKD-MBD) [43]

- Anemia [44] and malnutrition [31]

- Electrolytes [26,27]

- Volume [23]

B. Use of guideline-recommended therapeutic agents

- Use of medications [2,27,30]

C. Attainment of therapeutic targets

- Management of blood pressure [45]

- Glycemia [13,46,47]

- Lipids [48]

Nine quality indicators from four themes were commonly reported (in at least five studies). These common quality indicators included testing or monitoring of urine protein, serum creatinine/eGFR; treatment with ACEIs/ARBs and statins, avoidance of NSAIDs; management of blood pressure (with a target of ≤130/80 or ≤140/90, or without specific targets); and monitoring for HbA1c. For some themes, studies used various quality indicators for the same theme. For example, some studies applied various targets for blood pressure management while others did not. Also, different quality indicators for HbA1c and fasting glucose were used to monitor glycemia. The median number of quality indicators reported in each study was 7, ranging from 3 to 15; and there were 1 to 8 themes per study (the median was 3; S3 File).

## Quality of CKD management

Among the frequently reported quality indicators, there was significant variability in the median of the proportions of study participants meeting each indicator. For blood pressure control (≤130/80 mmHg), the median proportion was 39.5%, while for HbA1c monitoring, the median was significantly higher at 89.5%. Regarding kidney marker monitoring, the median proportion for serum creatinine/eGFR monitoring was 83.8%, compared to 40.4% for urine protein monitoring. For the use of medications, the median proportions were 44.5% for statins, 62.2% for ACEIs/ARBs, and 89.3% for non-NSAID medications. For blood pressure control, the median proportion varied from 39.5% for the ≤130/80 mmHg target to 75.7% with no target. Monitoring for HbA1c showed a significantly higher median proportion at 89.5%. The less commonly reported quality indicators appeared in themes of CKD-MBD, anemia and malnutrition, electrolytes, volume, and lipids (Table 3 in S3 Table). The presence of diabetes, hypertension, cardiovascular disease, or proteinuria was associated with higher achievement of quality indicators in monitoring of kidney markers, ACEIs/ARBs use, and management of blood pressure (Table 4 in S4 Table).

## Discussion

In this scoping review of 24 studies assessing the quality of CKD care using process quality indicators, 30 quality indicators were identified and categorized into three categories with nine themes, which were based on recommendations from international and national guidelines, as well as research studies. There are three key findings from this review. First, there is a lack of consistent reporting of key quality indicators and the evaluation of the quality of CKD management. Existing studies have applied various quality indicators that arise from different sources and covered different numbers and content of themes. Second, commonly reported quality indicators included the monitoring of urine protein, serum creatinine/eGFR; use of recommended medications (ACEIs/ARBs, statins, avoidance of NSAIDs), blood pressure, and HbA1c. There was limited information to assess the quality of CKD management for themes of CKD-MBD, anemia and malnutrition, electrolytes, volume, and lipids. Third, the management of CKD varied according to quality indicators, with satisfactory performance in monitoring of kidney function, and avoidance of NSAIDs, and the achievement rates tend to be higher among patients with diabetes, hypertension, cardiovascular disease, or proteinuria.

This review identifies the quality indicators covering various areas of CKD care but focusing on different themes. Consistent with current guidelines recommendations on management of progression and comorbid conditions of CKD [2,49], the quality of CKD management has been commonly assessed by indicators from the four themes: monitoring of kidney markers (urine protein and serum creatinine/eGFR); medications (ACEIs/ARBs, statins, and avoidance of NSAIDs); management of blood pressure; and monitoring for glycemia. These themes are covered by many studies and are important for CKD care, because inadequate monitoring and

treatment of comorbid conditions and use of nephrotoxic drugs may result in increased risks of disease progression and complications [50,51]. In contrast, quality indicators on prevention and management of CKD specific complications (e.g., bone disease, malnutrition, anemia) were assessed less frequently. These parameters are largely monitored in CKD clinics by nephrologists and/or internists, whilst the former ones predominantly focus on management of earlier CKD stages in primary care settings.

The performance of the quality indicators depends on the population under study, which was heterogeneous across studies in terms of causes and severity of CKD, prevalence of comorbidity conditions, and settings of healthcare. This is evidenced by the findings that generally, a higher achievement was observed in monitoring of kidney markers, the use of ACEIs/ARBs medications, and management of blood pressure in subgroups with diabetes, hypertension, cardiovascular disease, or proteinuria. However, results from some studies are below the desired target even among those with comorbidities [13]. This may be related to many different factors, such as patient-related factors including poor medication adherence, drug-drug interactions, patient's complexity and treatment priorities, the overall burden of medical care, and limited lifespan benefit [13,52]. There may also be provider-related factors, such as providers' knowledge, awareness, and skills regarding CKD management, as well as awareness of and adherence to guidelines, may influence the effectiveness of care delivery. Moreover, providers often face time constraints and conflicting demands (multiple other guidelines to contend with for other chronic conditions, like COPD, heart failure, etc.), which may impact their ability to prioritize and provide CKD care appropriately [53]. Finally, health system-related factors may play a role in CKD care quality. Access to recommended tests and medications, as well as limitations in time and resources, may influence the delivery of comprehensive and timely care for CKD patients [54].

As compared to monitoring of serum creatinine or eGFR, more studies have examined the testing for urine protein, but fewer have achieved the target for proteinuria measurement. There is strong evidence that the presence of albuminuria is associated with CKD progression and adverse events [2,55], and levels of albuminuria were inversely associated with recovery of kidney function [56]. While there is an increasing emphasis on albuminuria measurement for individuals at higher risk of progression, fewer studies have met this target in at least 75% of their study populations. Previous studies suggest that associations for not receiving this test included older age and rural location of residency [13]. Since older individuals tend to have a higher risk of death but a lower risk of kidney failure [57], this calls for a more individualized approach to monitoring albuminuria. The comprehensive monitoring of albuminuria and other CKD biomarkers has been advocated by the 2022 KDIGO controversies conference on improving CKD quality of care [58].

Diabetes and hypertension are the two most common causes of CKD worldwide, underscoring the importance of blood pressure management and blood glucose assessment in this patient population [59,60]. Results related to blood pressure quality indicators show that with a more stringent target ($\leq$140/90 or $\leq$130/80 mmHg), fewer studies achieved the target in at least 75% of their study participants, suggesting a room for improvement in this quality indicator. While controlling blood glucose levels as well as blood pressure can slow the progression of diabetic kidney disease [61], only four studies examined glycemia and three of them had an HbA1c test in at least 75% of study participants.

This study highlights a set of common quality indicators and the quality of CKD management varying with quality indicators. Gaps in CKD management raise awareness among healthcare providers to support meaningful changes to improve the quality of CKD care. Both qualitative and quantitative data are needed to differentiate the appropriate variance and inappropriate (low-quality) care to inform future quality improvement initiatives. A more

consistent and complete reporting of quality indicators is required for future studies assessing CKD care quality. Furthermore, current process quality indicators of CKD care have focused on physical health. Future studies should measure quality indicators of supportive care, which includes psychological, social, family, cultural, and spiritual support, as this is imperative for individuals with CKD, especially older adults, who are predominantly affected by this condition.

This study has limitations. First, the lack of standardized quality indicators poses a challenge to the analysis and may lead to inconsistencies in assessing the quality of CKD care across different settings or populations. In addition, for most quality indicators, only a small number of studies assessed their performance, making it difficult to assess the quality of CKD care in those themes. For example, while SGLT2 inhibitors have emerged as a key disease-modifying therapy to reduce proteinuria and delay CKD progression [62,63], only one study assessed SGLT2 inhibitors, which were used in less than 12% of the study population [36]. Second, findings from this review may not truly reflect the status of CKD management in practice, due to variations in study design, conduct, study population, sources of quality indicators, criteria used to evaluate these indicators, and reporting of quality indicators across studies. We acknowledge the possibility that some countries (regions) may deliver high-quality CKD care, yet data pertaining to their excellent practices may remain unpublished. Third, most studies included in this review were conducted in North America and Europe, and thus the findings from this review may not be generalizable to other regions, particularly in settings with limited economic resources. Our study findings have other generalizability considerations. While CKD occurs among children, pediatric CKD has unique aspects and thus its quality indicators deserve a separate examination [64].

In summary, existing studies have applied a variety of quality indicators that arise from different sources and cover a diverse number and content of themes. The quality of CKD care varies according to quality indicators. Consistent and complete reporting of quality indicators are warranted in future studies in this area.

## Supporting information

**S1 Fig. Fig 1.** Flow chart of selection of the included studies.
(TIF)

**S1 Table. Table 1.** Characteristics of included studies.
(DOCX)

**S2 Table. Table 2.** Extracted quality indicators and relevant themes.
(DOCX)

**S3 Table. Table 3.** Proportion of people with CKD meeting selected quality indicators in included studies.
(DOCX)

**S4 Table. Table 4.** Proportion of people with CKD meeting quality indicators in included studies, by comorbidity status.
(DOCX)

**S1 File. Preferred Reporting Items for Systematic reviews and Meta-Analyses extension for Scoping Reviews (PRISMA-ScR) checklist.**
(DOCX)

**S2 File. Search strategy.**
(DOCX)

**S3 File. Characteristics of included studies.**
(DOCX)

## Author Contributions

**Conceptualization:** Na Zhou, Zhaolan Yu, Yanhua Chen, Ping Liu.

**Data curation:** Na Zhou, Chengchuan Chen, Yubei Liu.

**Methodology:** Ping Liu.

**Writing – original draft:** Na Zhou, Chengchuan Chen.

**Writing – review & editing:** Aminu K. Bello, Yanhua Chen, Ping Liu.

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
