## [Decision Letter · Decision Letter 0]

16 Jun 2024

PONE-D-24-10881Assessing the quality of CKD care using process quality indicators: A scoping reviewPLOS ONE

Dear Dr. Liu,

Thank you for submitting your manuscript to PLOS ONE. After careful consideration, we feel that it has merit but does not fully meet PLOS ONE’s publication criteria as it currently stands. Therefore, we invite you to submit a revised version of the manuscript that addresses the points raised during the review process.

We look forward to receiving your revised manuscript.

Kind regards,

Ankur Shah

Academic Editor

PLOS ONE

Journal Requirements:

Additional Editor Comments :

The reviewers highlight limitations raised by the authors and offer a more direct alternative, please note their comments.

Reviewers' comments:

Reviewer's Responses to Questions

**Comments to the Author**

1. Is the manuscript technically sound, and do the data support the conclusions?

Reviewer #1: Yes

Reviewer #2: Yes

Reviewer #3: Partly

2. Has the statistical analysis been performed appropriately and rigorously? 

Reviewer #1: Yes

Reviewer #2: N/A

Reviewer #3: I Don't Know

3. Have the authors made all data underlying the findings in their manuscript fully available?

Reviewer #1: Yes

Reviewer #2: Yes

Reviewer #3: Yes

4. Is the manuscript presented in an intelligible fashion and written in standard English?

Reviewer #1: Yes

Reviewer #2: Yes

Reviewer #3: Yes

5. Review Comments to the Author

Reviewer #1: congratulation on the study, the study will have a great impact on the CKD management and treatment modality. and the study report can serve as a guideline for management of CKD.

the study was done comprehensively using PRISMA-ScR reporting items.

1. Regarding Eligibility criteria (18 years and older), as we know CKD is also prevalent among childrens too, the study could have been better since management plan differs for adult and children.

2. management and proper psychological counselling of patients regarding 'social and psychological health, KDQOL-SF' is also a quality indicator, which could be included in the study. (can add in the limitation of the study)

Reviewer #2: This was a well conducted scoping review. I am recommending the following improvements to improve the work but doubt they will make an appreciable difference in the resulting analysis.

1) The search was last run almost a year ago. It could be re-run to retrieve the most recent results to improve the timeliness of this review (for instance the PubMed search now retrieves about 100 additional results than what was originally searched).

2) Page 6 indicates that PRISMA-ScR is used as a methodology; this is slightly inaccurate as it is a reporting standard (see Sarkis-Onofre, R., Catalá-López, F., Aromataris, E. et al. How to properly use the PRISMA Statement. Syst Rev 10, 117 (2021). https://doi.org/10.1186/s13643-021-01671-z). You could re-word to indicate that you are reporting your methods in accordance with PRISMA guidelines. If you wish to consult or reference a scoping review methodology, you could review this chapter: Peters MDJ, Godfrey C, McInerney P, Munn Z, Tricco AC, Khalil, H. Scoping Reviews (2020). Aromataris E, Lockwood C, Porritt K, Pilla B, Jordan Z, editors. JBI Manual for Evidence Synthesis. JBI; 2024. Available from: https://synthesismanual.jbi.global. https://doi.org/10.46658/JBIMES-24-09

3) Page 6 line 102 should be Preferred Reporting Items for Systematic reviews and Meta-Analyses.

4) Regarding the search - if you have access to Embase or CINAHL databases, they would be valuable to search in addition to the selected databases.

5) Though not specified in the PRISMA-ScR, you could indicate what entitlements are included with your institutional Web of Science as this can vary between organizations.

6) What program or software was used for screening the articles?

7) Regarding the PRISMA flow diagram (Figure 1) - The total number of search results do not add up (the reported search numbers from each database total 11,308, not 11,271 as reported).

8) The Open Science Framework registration is not publicly available.

Reviewer #3: I appreciate the opportunity to review this original manuscript which provides a scoping review of quality indicators for chronic kidney disease (CKD) management from published studies to attempt to define a set of common quality indicators and performance on these indicators. The objectives of the scoping review are clearly stated, and the authors are addressing a gap in CKD management literature.

Summary and Major Issues:

The author's approach to evaluation of achievement of a quality indicator by using reference value of 75% for achievement in at least 3/4 of each original study population seems capricious and not a fair benchmark across all parameters. The authors acknowledge this themselves on page 20, lines 265-268, when they state "For these reasons, it may be inappropriate to determine the achievement of a specific quality indicator using uniform reference values across studies (e.g., achieving in at least 75% of study population). In my opinion, this approach is flawed as they state. I recommend a major revision of the manuscript to remove this arbitrary distinction of achieved/not achieved and instead report on the actual % of patients in each study who meet each quality indicator in the three categories/nine themes identified.

Minor Issues:

Several issues with readability or grammatical errors – specific instances noted are listed below (page numbers refer to manuscript pages):

1. Page 6, lines 108-109, the phrase “…which remain a lack of standardized quality of care assessment methodology and complete reporting.” should be rewritten for clarity.

2. Page 10, line 198 contains a typo – “stains” should be “statins.”

3. Most references to “non-NSAIDs” should perhaps be changed to “avoidance of NSAIDs.”

4. Page 11, line 210 “none studies” should be “no studies.”

5. Page 11, line 215 “statins use” should be “statin use.”

6. Page 13, line 254-255 “These themes are covered by many studies and important for CKD care…” should read “These themes are covered by many studies and are important for CKD care…”

7. Page 15, line 292 “…fewer studies were met in at least…” – “studies” seems like the incorrect word choice; needs clarification or revision

8. Page 17, line 323 contains grammatical error

9. Page 17, line 328 “jurisdictions” is incorrect word choice based on its definition

I recommend revision to address issues above to make this contribution to CKD management literature more relevant and readable. Thank you again for the opportunity to review.

6. PLOS authors have the option to publish the peer review history of their article (what does this mean?). If published, this will include your full peer review and any attached files.

Reviewer #1: **Yes: **Tshering Namgay

Reviewer #2: No

Reviewer #3: No

---

## [Author Response · Author response to Decision Letter 0]

17 Jul 2024

Reviewer #1: congratulation on the study, the study will have a great impact on the CKD management and treatment modality. and the study report can serve as a guideline for management of CKD.

the study was done comprehensively using PRISMA-ScR reporting items.

1. Regarding Eligibility criteria (18 years and older), as we know CKD is also prevalent among childrens too, the study could have been better since management plan differs for adult and children.

Response: Thank you for this suggestion. Pediatric CKD has unique aspects in terms of care delivery, growth, development, nutrition, long-term risks, and the transition to adult care (reference: KDIGO 2024 Clinical Practice Guideline for the Evaluation and Management of Chronic Kidney Disease). We believe that quality indicators of pediatric CKD care should be examined in a separate study. We have acknowledged this in the Limitation section on page 17: “…, particularly in settings with limited economic resources. Our study findings have other generalizability considerations. While CKD occurs among children, pediatric CKD has unique aspects and thus its quality indicators deserve a separate examination.”

2. Management and proper psychological counselling of patients regarding 'social and psychological health, KDQOL-SF' is also a quality indicator, which could be included in the study.

Response: Health Related QoL is generally considered as a patient-reported outcome. This review focused on process quality indicators. However, we agree with the reviewer that current process quality indicators of CKD care have focused on physical health, and supportive care (including psychological, social, family, cultural, and spiritual support) is imperative for individuals with CKD, especially older adults, who are predominantly affected by CKD.

We have added this to the Research Implication section on page 17: “A more consistent and complete reporting of quality indicators are required for future studies assessing CKD care quality. Furthermore, current process quality indicators of CKD care have focused on physical health. Future studies should measure quality indicators of supportive care, which includes psychological, social, family, cultural, and spiritual support, as this is imperative for individuals with CKD, especially older adults, who are predominantly affected by this condition.”

Reviewer #2: This was a well conducted scoping review. I am recommending the following improvements to improve the work but doubt they will make an appreciable difference in the resulting analysis.

1) The search was last run almost a year ago. It could be re-run to retrieve the most recent results to improve the timeliness of this review (for instance the PubMed search now retrieves about 100 additional results than what was originally searched).

Response: Following the reviewer’s suggestion, we have updated the literature search to June 20, 2024, and have included one more study meeting the eligibility criteria in this review.

2) Page 6 indicates that PRISMA-ScR is used as a methodology; this is slightly inaccurate as it is a reporting standard (see Sarkis-Onofre, R., Catalá-López, F., Aromataris, E. et al. How to properly use the PRISMA Statement. Syst Rev 10, 117 (2021).https://doi.org/10.1186/s13643-021-01671-z). You could re-word to indicate that you are reporting your methods in accordance with PRISMA guidelines. If you wish to consult or reference a scoping review methodology, you could review this chapter: Peters MDJ, Godfrey C, McInerney P, Munn Z, Tricco AC, Khalil, H. Scoping Reviews (2020). Aromataris E, Lockwood C, Porritt K, Pilla B, Jordan Z, editors. JBI Manual for Evidence Synthesis. JBI; 2024. Available from: https://synthesismanual.jbi.global. https://doi.org/10.46658/JBIMES-24-09

Response: We have re-worded to indicate that “We reported this scoping review according to the Preferred Reporting Items for Systematic Reviews and Meta-Analyses extension for Scoping Reviews (PRISMA-ScR).”

3) Page 6 line 102 should be Preferred Reporting Items for Systematic reviews and Meta-Analyses.

Response: We have corrected it. Thank you.

4) Regarding the search - if you have access to Embase or CINAHL databases, they would be valuable to search in addition to the selected databases.

Response: We have added the CINAHL database in our study search.

5) Though not specified in the PRISMA-ScR, you could indicate what entitlements are included with your institutional Web of Science as this can vary between organizations.

Response: Thank you. Our institution subscribes to the Web of Science Core Collection. We have added this in the Methods to improve transparency about the tools and data available for our research.

6) What program or software was used for screening the articles?

Response: We have added to the manuscript that NoteExpress software (version 3.7.0) was used for screening the identified articles.

7) Regarding the PRISMA flow diagram (Figure 1) - The total number of search results do not add up (the reported search numbers from each database total 11,308, not 11,271 as reported).

Response: We apologize for this error, as we did not initially include the number of imported Cochrane reviews. We have now revised the PRISMA flow diagram according to our updated search results.

8) The Open Science Framework registration is not publicly available.

Response: The Open Science Framework registration is now publicly available.

Reviewer #3: I appreciate the opportunity to review this original manuscript which provides a scoping review of quality indicators for chronic kidney disease (CKD) management from published studies to attempt to define a set of common quality indicators and performance on these indicators. The objectives of the scoping review are clearly stated, and the authors are addressing a gap in CKD management literature.

Summary and Major Issues:

The author's approach to evaluation of achievement of a quality indicator by using reference value of 75% for achievement in at least 3/4 of each original study population seems capricious and not a fair benchmark across all parameters. The authors acknowledge this themselves on page 20, lines 265-268, when they state "For these reasons, it may be inappropriate to determine the achievement of a specific quality indicator using uniform reference values across studies (e.g., achieving in at least 75% of study population). In my opinion, this approach is flawed as they state. I recommend a major revision of the manuscript to remove this arbitrary distinction of achieved/not achieved and instead report on the actual % of patients in each study who meet each quality indicator in the three categories/nine themes identified.

Response: We made a major revision of the manuscript to address this concern. We removed the arbitrary distinction of achieved or not achieved based on the 75% benchmark. Instead, we used the median and interquartile range (IQR) to summarize the distribution of proportions of study participants meeting a quality indicator across the included studies.

Minor Issues:

Several issues with readability or grammatical errors – specific instances noted are listed below (page numbers refer to manuscript pages):

1. Page 6, lines 108-109, the phrase “…which remain a lack of standardized quality of care assessment methodology and complete reporting.” should be rewritten for clarity.

Response: We have revised this sentence to read “We chose this approach because there is potentially a diverse body of literature assessing quality indicators for CKD management, and there is a lack of standardized quality of care assessment methodologies and complete reporting.”

2. Page 10, line 198 contains a typo – “stains” should be “statins.”

Response: We apologize for this and have made the correction.

3. Most references to “non-NSAIDs” should perhaps be changed to “avoidance of NSAIDs.”

Response: We have made the correction.

4. Page 11, line 210 “none studies” should be “no studies.”

Response: We have revised the paragraph.

5. Page 11, line 215 “statins use” should be “statin use.”

Response: We have revised the paragraph.

6. Page 13, line 254-255 “These themes are covered by many studies and important for CKD care…” should read “These themes are covered by many studies and are important for CKD care…”

Response: We have corrected the sentence as suggested.

7. Page 15, line 292 “…fewer studies were met in at least…” – “studies” seems like the incorrect word choice; needs clarification or revision

Response: We have changed that sentence to read: “While there is an increasing emphasis on albuminuria measurement for individuals at higher risk of progression, fewer studies have met this target in at least 75% of their study populations.”

8. Page 17, line 323 contains grammatical error

Response: Thank you for pointing this out. We have corrected the error and now it reads: “only one study assessed SGLT2 inhibitors, which were used in less than 12% of the study population”.

9. Page 17, line 328 “jurisdictions” is incorrect word choice based on its definition

Response: Thank you for the suggestion. We have corrected it to “countries (regions)”.

I recommend revision to address issues above to make this contribution to CKD management literature more relevant and readable. Thank you again for the opportunity to review.

---

## [Decision Letter · Decision Letter 1]

12 Aug 2024

Assessing the quality of CKD care using process quality indicators: A scoping review

PONE-D-24-10881R1

Dear Dr. Liu,

We’re pleased to inform you that your manuscript has been judged scientifically suitable for publication and will be formally accepted for publication once it meets all outstanding technical requirements.

Kind regards,

Ankur Shah

Academic Editor

PLOS ONE

Additional Editor Comments (optional):

Reviewers' comments:

Reviewer's Responses to Questions

**Comments to the Author**

1. If the authors have adequately addressed your comments raised in a previous round of review and you feel that this manuscript is now acceptable for publication, you may indicate that here to bypass the “Comments to the Author” section, enter your conflict of interest statement in the “Confidential to Editor” section, and submit your "Accept" recommendation.

Reviewer #2: All comments have been addressed

Reviewer #3: All comments have been addressed

2. Is the manuscript technically sound, and do the data support the conclusions?

Reviewer #2: Yes

Reviewer #3: (No Response)

3. Has the statistical analysis been performed appropriately and rigorously? 

Reviewer #2: N/A

Reviewer #3: (No Response)

4. Have the authors made all data underlying the findings in their manuscript fully available?

Reviewer #2: Yes

Reviewer #3: (No Response)

5. Is the manuscript presented in an intelligible fashion and written in standard English?

Reviewer #2: Yes

Reviewer #3: (No Response)

6. Review Comments to the Author

Reviewer #2: Thank you for addressing my concerns regarding updating your search as well as adding an additional database for breadth of coverage.

Reviewer #3: The authors have addressed my comments from the initial draft satisfactorily. The manuscript reads well and presents a cohesive scoping review of CKD quality indicators.

7. PLOS authors have the option to publish the peer review history of their article (what does this mean?). If published, this will include your full peer review and any attached files.

Reviewer #2: **Yes: **Kelsey Sawyer MS

Reviewer #3: No

---

## [Editor Report · Acceptance letter]

23 Aug 2024

PONE-D-24-10881R1 

PLOS ONE

Dear Dr. Liu, 

I'm pleased to inform you that your manuscript has been deemed suitable for publication in PLOS ONE. Congratulations! Your manuscript is now being handed over to our production team.

Kind regards, 

on behalf of

Dr. Ankur Shah 

Academic Editor

PLOS ONE